# Rheological Properties of Lunar Mortars

**Joanna J. Sokołowska** *, **Piotr Woyciechowski**  and **Maciej Kalinowski** 

Department of Building Materials, Faculty of Civil Engineering, Warsaw University of Technology,
00-637 Warszawa, Poland; p.woyciechowski@il.pw.edu.pl (P.W.); maciej.kalinowski.dokt@pw.edu.pl (M.K.)
* Correspondence: j.sokolowska@il.pw.edu.pl; Tel.: +48-22-2346482

**Abstract:** NASA has revealed that they plan to resume manned missions and ensure the permanent presence of people in the so-called habitats on the Moon by 2024. Moon habitats are expected to be built using local resources—it is planned to use lunar regolith as aggregate in lunar concrete. Lunar concrete design requires a new approach in terms of both the production technology and the operating conditions significantly different from the Earth. Considering that more and more often it is assumed that the water present on the Moon in the form of ice might be used to maintain the base, but also to construct the base structure, the authors decided to investigate slightly more traditional composites than the recently promoted sulfur and polymer composites thermally hardened and cured. Numerous compositions of cement "lunar micro-mortars" and "lunar mortars" were made and tested to study rheological properties, namely, the consistency, which largely depend on the morphology of the fine-grained filler, i.e., regolith. For obvious reasons, the lunar regolith simulant (LRS) was used in place of the original Moon regolith. The used LRS mapped the grain size distribution and morphology of the real lunar regolith. It was created for the purpose of studying the erosive effect of dusty regolith fractions on the moving parts of lunar landers and other mechanical equipment; therefore, it simulated well the behavior of regolith particles in relation to cement paste. The obtained results made it possible to develop preliminary compositions for "lunar mortars" (possible to apply in, e.g., 3D concrete printing) and to prepare, test, and evaluate mortar properties in comparison to traditional quartz mortars (under the conditions of the Earth laboratory).

**Keywords:** lunar regolith simulant; LRS; lunar micro-mortars; lunar concrete-like materials; extraterrestrial construction materials; rheological properties

## 1. Introduction

On 21 July 1966, the first manned mission landed on the Moon (Apollo 11) [1]. In 2016, the 50th anniversary of this event was celebrated, which reawakened interest in the Moon and space travel. National Aeronautics and Space Administration (NASA) has revealed plans to resume manned missions to the Moon: mission Artemis III is planned to be the first crewed lunar landing since Apollo 17 in 1972—the crew is to land on the Moon by 2024, and by 2028, it is planned to implement sustainable exploration of the Moon [2–4]. Earlier, NASA and the European Space Agency announced that they wanted to ensure the possibility of permanent human residence in the so-called habitats on the Moon or Mars before 2040 [2]. Regardless of the time horizon, the implementation of the concept of extraterrestrial habitats requires the development of material and execution concepts of a lunar habitat.

In order to implement such an unusual construction project as a base/habitat on the Moon, many factors need to be considered, including advanced economical, technological, and material engineering aspects. The latter two are closely related and limited by the former, because the cost of transporting cargo to the Moon, although it has decreased in recent years, is still calculated in the tens of thousands of dollars. The price for one launch of a new NASA's heavy-lifting vehicle—the Space Launch System, designed for delivering payloads over 25 metric tons to the Moon, is estimated to be over 2 billion dollars [5]; this

corresponds to ca. USD 75,000/kg of delivery cost [6]. In 2020, NASA announced a tender for a transport service to the Moon (service delivery until 2023) of the VIPER mobile robot, which will search for water (ice) at one of the Moon's poles [4].

Taking into account the "astronomical" cost of transporting cargo, including equipment and materials, scientists for decades have been supporting the concept of using locally available resources to build a base/habitat, including the so-called lunar regolith—layers of dust and rock debris covering the lunar surface [7–9]. These plans received greater attention after the American Lunar Crater Observation and Sensing Satellite (LCROSS) detected a significant amount of water (5.6% by mass) in 15 tons of excavated regolith in 2009 [10].

Various research centers are attempting to develop concrete-like building composites that could potentially be made on the Moon. These composites are often called the term "lunar concrete" [11] and belong to the wider group of the so-called extraterrestrial construction materials [8]. Most of those assume the use of lunar regolith as aggregate, while in terms of the binder, mainly three groups of concrete can be distinguished: polymer concretes [11,12], sulfur concretes [13], and ordinary concretes (with ordinary Portland cement, OPC). The advantages and disadvantages of these material solutions in the context of their manufacture and use on the Moon, as well as the effects of the first attempts to prepare them in Earth laboratories (with the partial simulation of lunar conditions), are discussed later in the article. First, the conditions for the performance and operation of concrete on the Moon are specified, which are important assumptions for the design of durable lunar concrete.

The following section describes brief characteristics of the lunar regolith based on samples collected on the Moon [1] (landing and sampling sites are shown in Figure 1)—astronauts have collected and delivered to Earth a total of 382 kg of rock and lunar dust [14]. Particular attention was paid to the granulation of regolith because it was important from the point of view of the research part of the work presented in this paper.

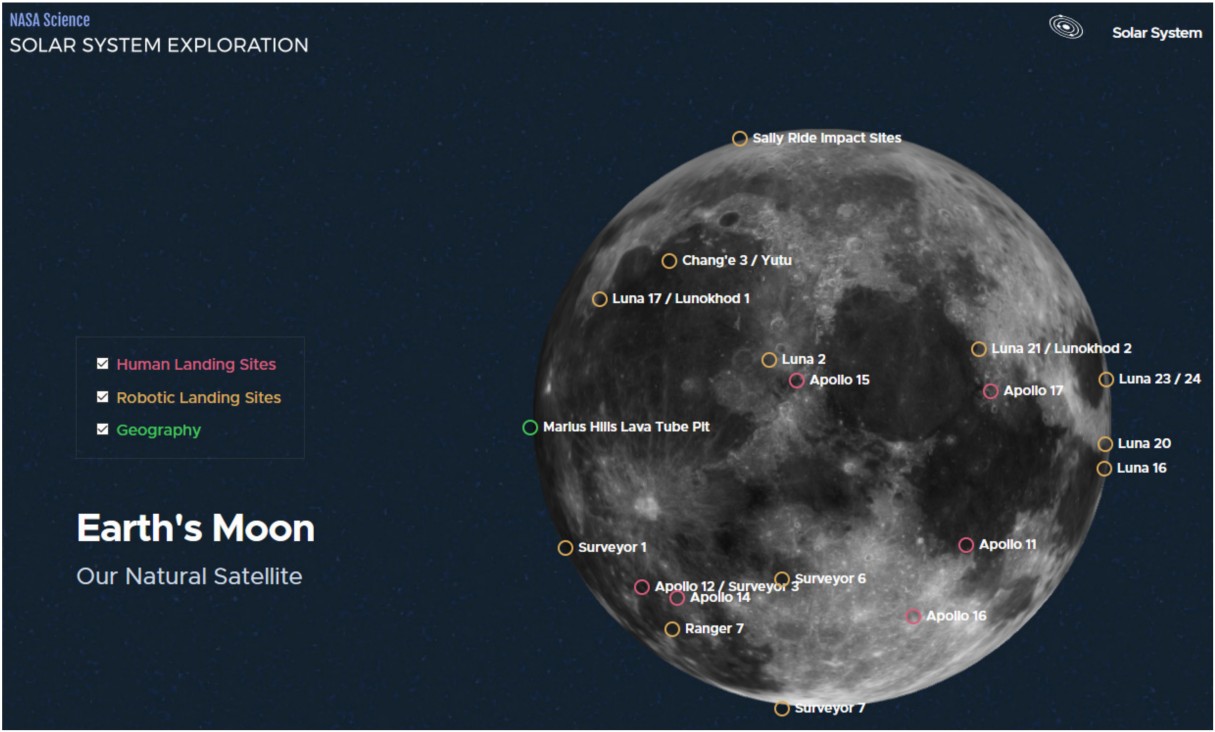

**Figure 1.** Moon seen from the Earth's side with the landing sites of manned and non-manned missions marked (there was no landing on the other, so-called dark side); screenshot from NASA interactive map [14].

The aim of the research was to initially assess the possibility of using regolith as aggregate in cement mortars. It has been studied indirectly because, for obvious reasons, it was not possible to use an original lunar regolith, and therefore, the lunar regolith simulant (LRS) that mapped the grain size distribution of the real regolith was used instead. The used lunar regolith simulant was created for the purpose of studying the erosive effect of dusty regolith fractions on the moving parts of lunar landers and other mechanical equipment; therefore, it simulated well the behavior of regolith particles in relation to cement paste. Additionally, the research itself was aimed at the rheological characterization of concrete mixes with the lunar regolith simulant, important from the point of view of the technology of making elements on the Moon. The obtained results made it possible to develop preliminary compositions for "lunar mortars" with w/c ratio possible to apply in e.g., 3D concrete printing and to prepare, test, and evaluate mortar properties in comparison to traditional quartz mortars (under the conditions of the Earth laboratory).

### 1.1. Material Characteristics of the Lunar Regolith

Regolith is the upper dust-like layer of lunar ground, with a thickness of around 4–15 m [15]. In terms of morphology, the lunar regolith consists of spherical and oblong grain dust, angular rock particles, and dendritic slag grains [16]. Already in the 1970s, it was stated that the particle size distribution (PSD) of the Moon soil is the basis for its classification [17]. The size of individual particles of lunar regolith delivered to Earth generally varies from few microns to hundreds of millimeters. Samples provided in subsequent Apollo missions (Apollo 11, 12, 14, and 15) enabled the development of some average particle size characteristics of the fine fraction of lunar regolith, pointing $d_{50}$ (the size point below which 50% of the tested material is contained., i.e., median) with values in the range of 42–130 μm (see Table 1).

**Table 1.** Lunar soil grain distribution based on the results obtained after missions Apollo 11, 12, 14, and 15 [17].

| Mission | Number of Samples | $d_{20}$, mm | $d_{50}$, mm | $d_{80}$, mm |
|---------|------------------|-------------|-------------|-------------|
| Apollo 11 | 13 | 0.016–0.030 | 0.048–0.105 | 0.163–0.720 |
| Apollo 12 | 55 | 0.013–0.033 | 0.042–0.094 | 0.167–0.440 |
| Apollo 14 | 24 | 0.013–0.036 | 0.044–0.130 | 0.061–0.200 |
| Apollo 15 | 19 | 0.014–0.019 | 0.051–0.108 | 0.180–0.400 |
| Altogether | 111 | 0.013–0.036 | 0.042–0.130 | 0.163–0.720 |

A more recent paper (published by NASA in 2007) [18] shows that the particle size distribution generally follows the log-normal curve with the mean values in the range of 45–100 μm. "Dust" fraction of lunar regolith includes particles smaller than 10 μm, of which about 90% (by weight) are particles smaller than 1 μm [19], although some particles can be even as small as 10 nm [18]. Particles with dimensions larger than 0.25 mm (250 μm) constitute 10% (by weight) of regolith [20]. Moreover, lunar regolith contains morphological forms that do not exist on Earth—these are spherical lunar chondrules (with dimensions from a few microns to 0.5 mm) formed as a result of meteorite falls and the sudden melting of lunar rocks) [19]. The fragmentation of the lunar regolith is the result of mechanical impacts but also thermal stress caused by high daily temperature differences and erosion caused, among others, by ionizing radiation of the solar wind and galactic cosmic radiation [21]. Figure 2 shows the averaged chemical composition of regolith and the graining curve and an example of an SEM micrograph [20] of lunar dust.

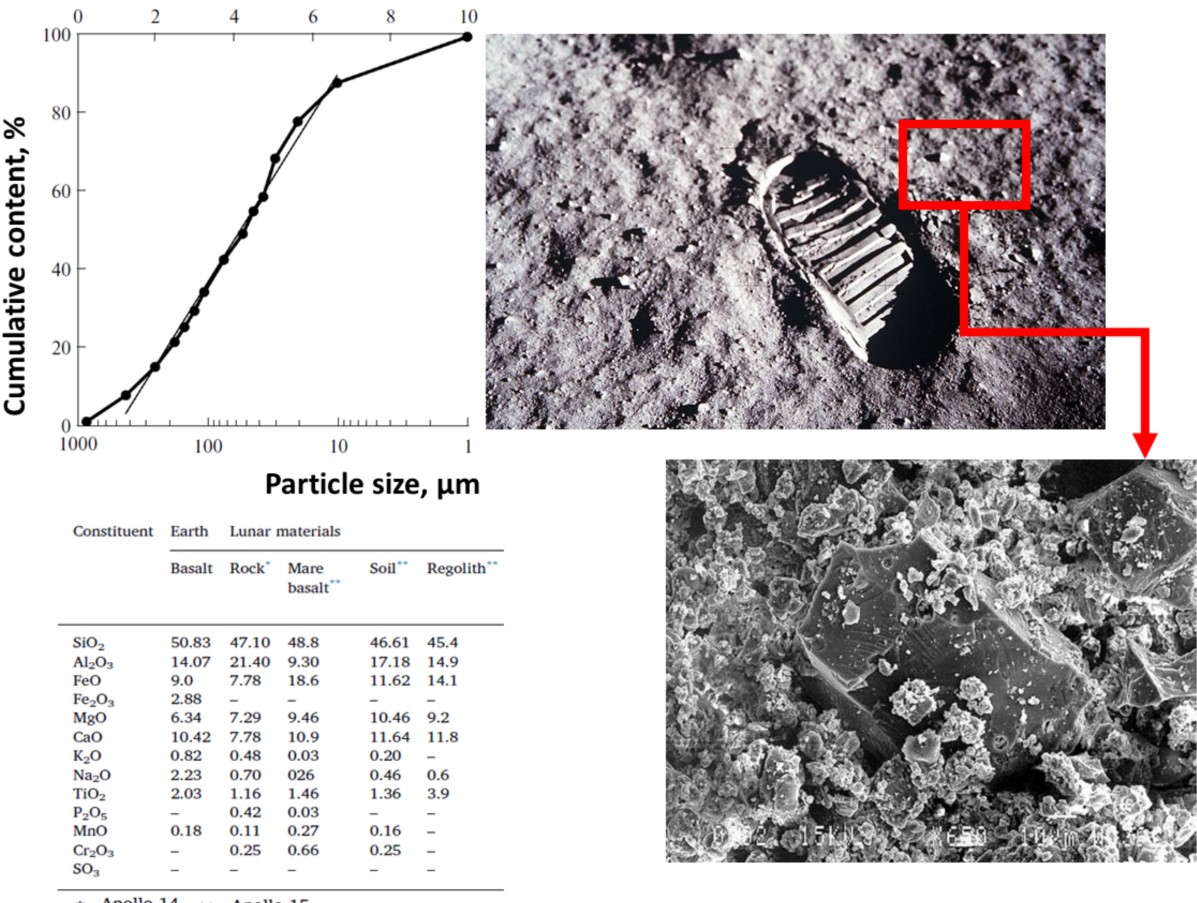

**Figure 2.** Morphology, particle size distribution (PSD) plot, and chemical composition of the lunar regolith (based on [16,19,20,22]).

Lunar regolith is characterized by very strong adhesion to various surfaces. This is a negative property in the context of functioning on the surface of the Moon. Regolith adheres to the surface of astronauts' suits and to the surface of machines. Adhesion in combination with the high abrasiveness of regolith can lead to damage to machinery mechanisms, research equipment, including optics. However, in the context of adhesion of binder to aggregate in cementitious materials, this feature seems to be very important and desirable.

*1.2. Conditions of Concrete Preparation and Performance on the Moon*

The conditions on the Moon differ significantly from those on Earth, which largely determines the choice of components and technology of execution and use of elements and structures made of lunar concrete. The most important differences are the lack of an atmosphere, less gravity, and scarce water resources. A very important issue is the large temperature fluctuations on the surface of the Moon in the circadian cycle (which lasts 656 h).

The temperature on the surface of the Moon varies greatly—at the equator, it ranges from −180 °C to +123 °C [23]. Locations from the side of the Earth (on the bright side) and closer to the equator are indicated as optimal due to the temperature criterion (and the possibility of obtaining energy from photovoltaic panels). Based on the trajectories of flights from Earth's previous missions, including manned missions, landed there (Figure 1), the poles and the other side of the Moon are excluded as habitat locations as the conditions there are extremely unsuitable for humans.

The scarcity of liquid water on the Moon adds difficulty in carrying out construction work, especially in terms of the potential use of cement concrete. This applies to both the use of water as a component of concrete and for its subsequent maintenance, as well as in

work related to the cleaning and maintenance of equipment and molds. Therefore, concrete-like composites with water-free organic synthetic polymer binders or sulfur binders are also considered types of lunar concrete [13,23].

Considering the aforementioned low temperatures at poles and the fact that the vicinity of poles is not destined for the landing of terrestrial landers, as well as for the construction of the base itself, the search for water (in the form of ice) requires the use of complex technical solutions, reliable at very low temperatures and water/ice transport over relatively long distances. The cost of carrying out such a water extraction process is currently disproportionate to its efficiency. However, NASA plans to search for water ice with the aforementioned VIPER rover so that it would not be necessary to transport water from Earth.

The gravity on the Moon is six times smaller than on Earth (i.e., $1.62 \ m/s^2$) [19]; thus, the technology of performing construction work on the Moon requires adaptation to the reduced gravity. Horizontal and vertical transport can be performed with less energy, while obtaining tight piles of crumb aggregate and proper mixing and compaction of the mixture may be difficult. As a result of such low gravity, segregation of the concrete mix may occur. Heat treatment with a closed-circuit heating factor might be a feasible method for developing concrete on the Moon, which has significant advantages over conventional concreting (faster and more intense hydration, resulting in shorter curing time). Additionally, the autoclaving method prevents the material from being exposed to a lunar vacuum and thus avoids rapid water evaporation [24].

The impact of the practical lack of atmosphere (atmospheric pressure on the Moon is approx. $3 \cdot 10^{-13}$ kPa) [19] on the maintenance and subsequent exploitation of composites should also be considered—both in the context of the absence of oxides that react with concrete components (e.g., carbon dioxide activity leading to concrete carbonation) [25] and in the context of exposure of concrete to solar radiation unfiltered by the atmosphere, including ionizing radiation and UV radiation, which are the cause of intense aging, especially for concrete containing polymers [12].

The mineralogical composition of the lunar regolith is dominated by anorthosite (mainly plagioclase with a small amount of pyroxenes), basalts, and glazes (enamel fragments and agglutinates)—as in the case of some terrestrial volcanic soils; however, quartz and clay minerals are absent. Chemically, lunar regolith is characterized by a high content of metallic iron as well as titanium and helium (including the $^3$He isotope). All these facts make lunar regolith an extremely chemically active material that can act as a catalyst for many reactions [16,24].

### 1.3. Lunar Regolith Simulants

A lunar regolith simulant is any material made from natural or synthetic earth or meteoric components to replicate one or more of the physical or chemical properties of the lunar regolith. It is impossible to produce a model of lunar soil on the basis of rocks on Earth that would reflect its physical, mechanical, electromagnetic, thermal properties, as well as the chemical or mineralogical composition. For this reason, all models and analogs of lunar regolith are produced to simulate one or two of the aforementioned characteristics necessary for research and experimentation. All stimulants are produced by mechanical rock crushing. In order to better reflect the particle size distribution and shape of the particles, the rocks are crushed by impacts. The granulometric composition of most simulants differs from the lunar regolith. They mainly correspond to coarser fractions since obtaining very fine particles is associated with a difficult and expensive technology. Basalt rocks or anorthosite with an admixture of olivine and pyroxene are usually used as the main components of lunar regolith analogs. In order to map agglutinates, glass is added to the mixture of rocks [16]. Over the years, several different lunar regolith stimulants were developed, including several variants of Johnson Space Center Number One (JSC-1), two lunar highlands simulants, namely, the Lunar Highlands Type (NU-LHT) series and Olivine Bytownite (OB-1), as well as OPRL2N Standard Representative Lunar Mare

Simulant, Standard Representative Lunar Highland Simulant, Minnesota Lunar Simulant 1 (MLS-1), Chenobi, etc. [26].

Polish scientists from the Space Research Center of the Polish Academy of Sciences, due to the need to use large amounts of lunar regolith simulant for research but also high prices of known lunar regolith simulants resulting from the difficult methods of production, decided to create their own simulant [27]. The aim of the invention was to produce material needed in research on assessment of the reliability of machines and tools working in simulated lunar-like conditions, i.e., in the presence of a large amount of abrasive dust. It was important to map the sharp-edged grains of lunar regolith minerals [28].

Generally, Polish lunar regolith simulant is a mix of components appearing in Polish geological conditions and/or commercial components produced by local raw mineral producers, including mechanically crushed quartz, granite sand, and grit. As the components were only mixed without additional technological procedures, the new LRS was obtained at a much lower cost and required a shorter production time.

Despite the simple approach to production, the Polish lunar regolith simulant is characterized by the grain size distribution, color, mechanical strength, cohesion, and an internal friction angle comparable to the Chenobi simulant. In Figure 3, one can compare the particle size distribution plots of Chenobi simulant and Polish lunar regolith simulant (marked AGK-2010). In terms of morphology, as required, it consists of full sharp-edged particles (mechanical fragmentation effect). It is mainly a gray noncohesive dusty powder of bulk density of (1.290–1.300) g/cm$^3$, an internal friction angle in the range of (36.5–38.5)°, and cohesion in the range of (3.8–4.1) kPa [26,27].

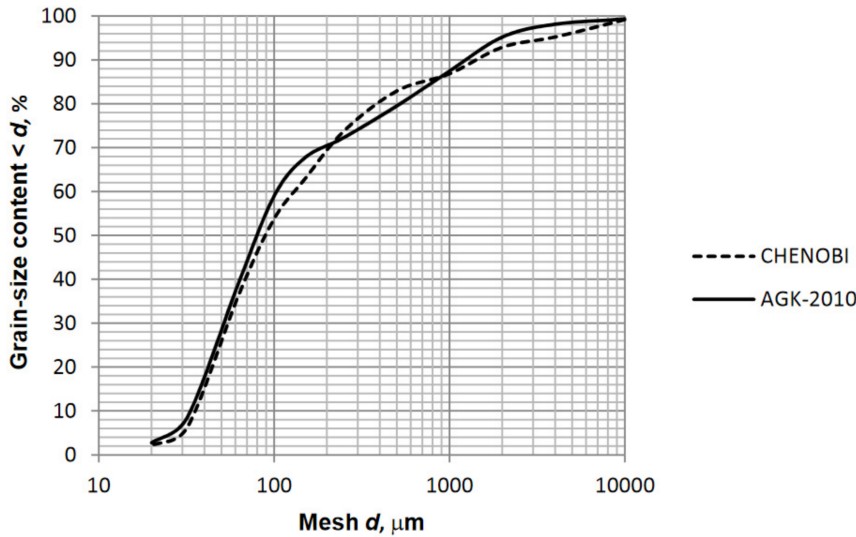

**Figure 3.** Particle size distribution (PSD) of the lunar regolith simulants—American "Chenobi" of uniformity coefficient, Cu = 3.68 (dashed line) and Polish "AGK-2010" of uniformity coefficient, Cu = 3.54 (solid line). Reprinted from [28].

The development of the Polish lunar regolith simulant was aimed at testing the KRET penetrator, testing the planetary rovers, promoting research related to space exploration in Poland, etc. [27]. This LRS was also used in the research considering the subject of this paper.

## 2. Research and Scope

It was very difficult to design the research program determining the potential use of lunar regolith. The research aim was to find a composition with the highest accepted lunar regolith simulant content in the mass of the composite while not worsening composite rheological properties. Therefore, the most difficult task was to decide on the initial quantitative composition of the tested composites so that the results could be interpreted and assessed. Considering the very fine particle size distribution of lunar regolith and

therefore very high water demand ratio, when replacing aggregate with lunar regolith in mortar, it was not possible to perform a simple substitution of sand with regolith simulant. If the substitution were to be carried out directly, the aggregate of comparable grain size should be replaced. For this reason, mortars containing both sand (aggregate up to 2 mm) and quartz powder were tested. However, in order to have any known and recognized reference point described in the standards, it was necessary to prepare a classic standard mortar (meeting the requirements of EN 196-1 standard), which then began to be modified.

Therefore, the adopted sequence of experimental stages began with the determination of material characteristics of Polish lunar regolith simulant (LRS) but also the selection of similarly graded quartz powder. The next stage was the experimental initial determination of the possible grade of filling the standard mortar with very fine filler (the so-called micro-filler: quartz powder or lunar regolith simulant) keeping the water/cement ratio identical to that of the standard mortar (i.e., w/c = 0.50) and with constant consistency (not to introduce additional variables apart from the aggregate composition). At the next stage, apart from the remaining sand, the interaction between the cement paste (water + cement in a constant proportion) phase and the LRS micro-filler phase was tested. At that stage, the micro-mortars with regolith simulant were tested and the influence of LRS on the rheological characteristics of such mixes was assessed. The important task was to obtain compositions of a possibly high content of LRS and good workability of the mixes (necessary to enable efficient filling of the molds with the lunar mortar mix) and on the other hand, reduce water content. At this stage, the analyzed input variable was the content of LRS in relation to cement mass, and the output data were the consistency of lunar micro-mortars tested with various methods. Additionally, the compressive strength and flexural strength of selected lunar mortars were tested after 28 days and compared to the properties of reference standard mortar. The subsequent stages of the experimental program are presented in Table 2.

**Table 2.** Stages of the experimental program (LRS—lunar regolith simulant; QP—quartz powder).

| Stage | Performed Tests | Obtained Effects |
|---|---|---|
| Stage 1: Micro-fillers (LRS and QP) characterization | • Density (Le Chatelier)<br>• Particle size distribution<br>• Specific surface area<br>• Morphology (micrographs) | • Characterization of LRS<br>• Selection of QP of similar granulometry (subject of substitution with LRS) |
| Stage 2: Initial determination of the content of micro-filler in the mortar aggregate | • Flow table test | • Determining the effect of max. share of QP in aggregate while maintaining w/c ratio and satisfactory workability of the mortar: 20% of aggregate mass (using large dose of superplasticizer) |
| Stage 3: "Lunar micro-mortars" consistence | • Standard consistency in Vicat apparatus<br>• Modified flow table test<br>• Flow table test | • Determining the effect of LRS on the "lunar micro-mortar" consistence (2 series: w/c = 0.265 and w/c = 0.500) |
| Stage 4: "Lunar mortars" mechanical properties | • Compressive strength<br>• Flexural strength | • Additional characterization of basic mechanical properties of selected "lunar mortars" and comparison to standard mortars properties |

Given the environmental "circumstances" on the Moon, it can be assumed that traditional concreting methods could not be applicable. However, 3D concrete printing technology presents an interesting option for lunar concrete construction [29]. From this point of view, consistency was considered one of the main properties that could affect the technological process of producing elements of cementitious composites.

The research on the evaluation of workability parameters in 3D concrete printing showed that the "flow test table was more consistent than the other methods and printability window for the printing system was found between flow values of 18 and 24 cm" [30]. In other research [31], it was stated that the "pumpability" of the cementitious mix is suitable for 3D printing if the flow value is between 150 and 190 mm and such a mix provides a smooth surface and high "buildability."

## 3. Materials and Methods

The tested micro-mortars consisted of water, cement, quartz powder, or regolith simulant. The tested mortars consisted of the abovementioned components, supplemented with fine aggregates of up to 2 mm. The explanation of why standard sand was used as aggregate is provided later in the chapter. The technical characteristics of the components used to prepare tested specimens are presented below.

For the mixing water in tested pastes and mortars, tap water was used, meeting the requirements of standard EN 1008. As a binder the CEM I 42.5 R, i.e., ordinary Portland cement of strength class 42.5 and high early strength (R), compliant with the EN 197-1 and EN 197-2 standards, was used. The basic technical properties of the cement declared by the manufacturer are given in Table 3.

**Table 3.** Physical and mechanical properties of cement.

| Physical and Mechanical Properties | EN 197-1 Requirements | Average Values |
|---|---|---|
| Beginning of setting time [min] | min. 60 | 192 |
| Compressive strength after 2 days | min. 20.0 | 29.1 |
| Compressive strength after 28 days | 42.5–62.5 | 55.5 |
| Specific surface area [$cm^2$/g] | - | 3816 |
| Water for standard consistency [%] | - | 27.0 |

As mentioned earlier, a search was made for a filler (or rather a micro-filler) that would constitute the aggregate fraction as similar as possible to lunar regolith in terms of granulation and morphology. Ultimately, a commercially available quartz filler (mechanically crushed quartz in powder form) was selected. The own test results showing the characterization of the chosen powder are given in the next section.

Standard sand meeting the requirements of EN 196-1 was used as fine aggregate. The use of standard sand was justified for two reasons. First, there was no intention to introduce additional material variables (and the lunar mortars were finally compared to the results of the standard mortar). Secondly, standard sand has a very low content of dusty particles—the standard allows a maximum of 2% of grains smaller than 0.08 mm. Therefore, with these particles, it was possible to very precisely dose micro-fillers (i.e., quartz powder or lunar regolith simulant).

As the aim of the research was to maximize the share of lunar regolith simulant, in some cases, it was necessary to use a superplasticizer. A very effective commercial superplasticizer (based on polycarboxylate ethers) was used, the recommended dosing of which was up to 1.5% in relation to the cement mass.

The density of the lunar regolith simulant and quartz powder was determined using the Le Chatelier volumenometer (consisting of a flat-bottomed flask) according to the procedure described in the EN 1936 standard. The presented values are the average of 3 results.

The particle size distribution (PSD) measurements were performed by the laser scattering method using the laser analyzer Horiba LA-300 (Kyoto, Japan). The test involved passing laser beams through a 0.1% sodium polymetaphosphate solution containing particles of tested material dispersed by ultrasounds and determining the particle size (in the range of 0.01–600 μm). The values of statistical parameters describing the micro-fillers particle size distributions, as well as specific surface area and abovementioned density are given in Table 4.

**Table 4.** Statistical parameters describing the particle size distribution, specific surface area, and density of regolith simulant (LRS) and quartz powder (QP).

| Parameter | LRS | QP |
|---|---|---|
| $D_{min}$ [μm] | 1.0 | <1.0 (0.584) |
| $D_{10}$ [μm] | 6.7 | 3.4 |
| $D_{50}$ (median) [μm] | 37.0 | 21.5 |
| $D_{av}$ (average) [μm] | 54.9 | 28.4 |
| $D_{90}$ [μm] | 133.1 | 67.5 |
| $D_{max}$ [μm] | 300.5 | 152.4 |
| Mode [μm] | 55.1 | 32 |
| Specific surface area [$cm^2/cm^3$] | 3981 | 7036 |
| Density [$g/cm^3$] | 2.641 ($C_V$ = 0.14%) | 2.650 |

A Nikon Eclipse E-200F PLUS fluorescence microscope with trinocular attachment and a digital camera were used to determine the lunar regolith simulant morphology. The proper micrograph is given in Figure 4.

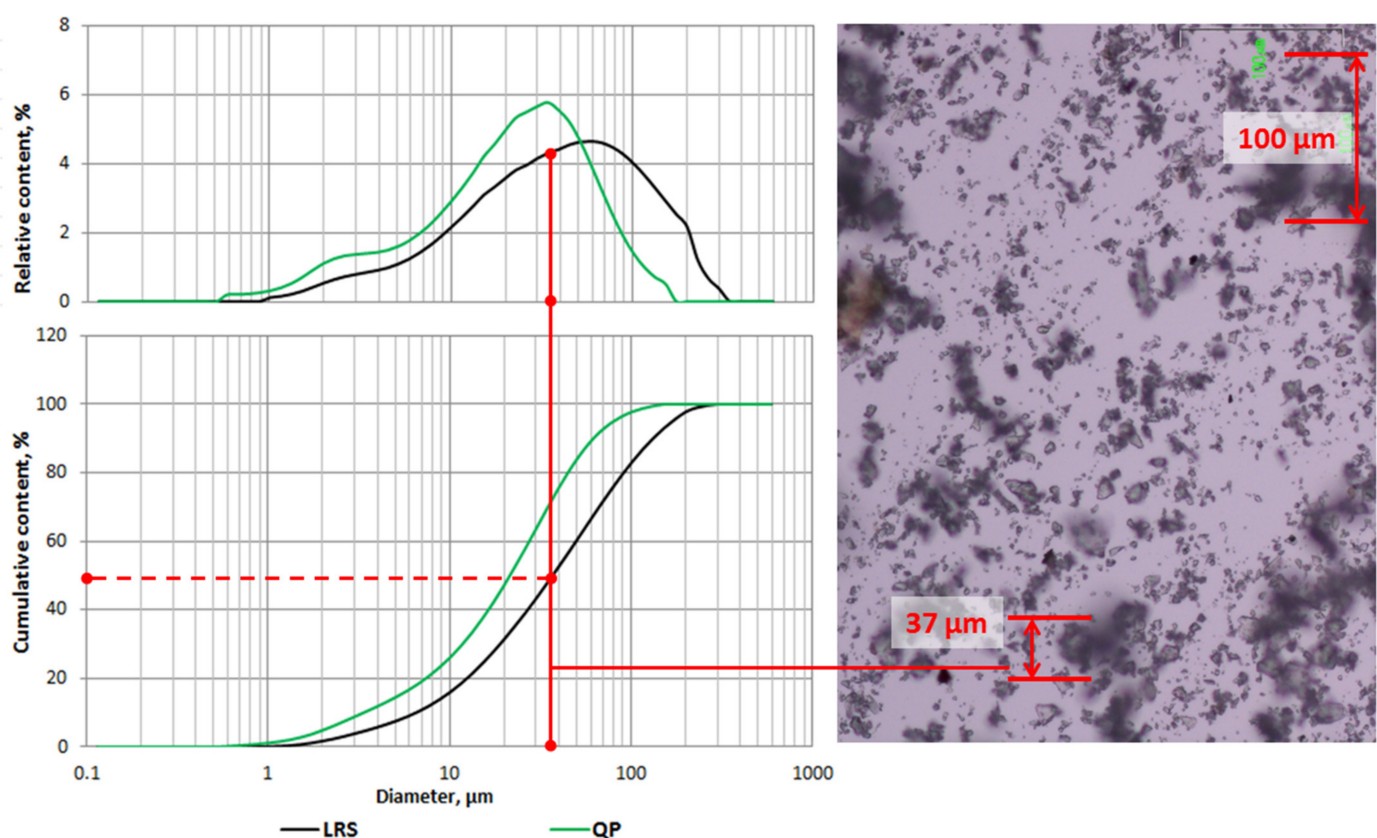

**Figure 4.** Particle size distribution plots of lunar regolith simulant (LRS) and quartz powder (QP) and micrograph of sharp-edged particles of LRS (magnification: 200×).

The consistency was tested using three methods. In the case of mortars (aggregate up to 2 mm), the consistency was tested with the standard flow table method (according to the procedure in EN 1015-3 standard). In the case of micro-mortars, the consistency was tested using Vicat apparatus, as in the case of standard consistency according to EN 196-3, and using a modified flow table method—developed particularly for the needs of

the experiment. The modification consisted of changing the mold filled with the tested mix—due to the different fluidity of the micro-mortars, the volume of the sample had to be reduced; therefore, the micro-mortar mix was placed on the flow table, not in a truncated cone ring but in the so-called Vicat ring (usually used to determine the standard consistency of the cement paste according to EN 196-3 standard). The remaining test parameters including the application of the sample in the ring, the number of flow table shakes, and the measurement of the flow diameter remained unchanged from the original flow table method.

Flexural strength and compressive strength were tested according to the method described in EN 196-1 standard using prism-shaped specimens with dimensions of 40 mm × 40 mm × 160 mm (and latter halves of those prisms remaining after the three-point bending test). The presented strength values are the average (of 3 or 6 results). Specimens were stored in the laboratory conditions for periods of 28 days.

## 4. Results and Discussion

### 4.1. Characterization of Lunar Regolith Simulant (LRS) and Quartz Powder (QP)

Figure 4 presents the particle size distribution plots (the relative frequency and the cumulative relative frequency of the particle size distribution) of lunar regolith simulant, LRS and quartz powder, QP, and the micrograph of the LRS. The values of statistical parameters describing the fillers' particle size distributions, as well as specific surface area (calculated from the distribution, making an assumption about the spherical shape of the particles) and density determined in a Le Chateliere volumenometer are given in Table 4.

The simple micrograph confirmed that the tested lunar regolith simulant is characterized by sharp-edged particles, as assumed during production, which enables a good simulation of the mixing of the components of the lunar micro-mortars and mortars. Already at this stage, it can be inferred that in relation to round and smooth grains of sand, the presence of regolith simulant would affect the consistency of mixes negatively. Water demand, i.e., the need for water to obtain a specific consistency of a mix, depends primarily on the size of the grains but is also influenced by the roughness and shape of the grains.

Taking into account the important role of the smallest fractions that determine the specific surface area (the finer the aggregate, the larger the total surface area) and the resulting water demand ratio, the focus was primarily on the finer fractions of the tested material. The results of the test performed on the laser analyzer showed that 16% of the LRS particles are smaller than 10 μm. The real lunar regolith consists of about 90% of particles larger than 10 μm [16]; therefore, it was confirmed that the simulating material is in agreement with the simulated lunar soil for particles < 10 μm. The median of LRS PDS is 37 μm, which is also in line with the real lunar regolith data (40 μm–130 μm) [16,17]. Additional analysis showed that the average size of RLS particles does not exceed 55 μm, while the largest registered size of the LRS is 300 μm. Despite the last result, it is worth mentioning that 90% of total lunar regolith simulants are not larger than 133 μm (less than half of the size of the biggest registered particle).

As for the density, taking into account that the tested material was a simulant obtained by grinding rocks rich in quartz (silica, granites from Polish deposits, which are mainly made of plagioclase, quartz, biotite, and hornblende [32]), it is not surprising that the experimentally obtained value of density of the LRS simulant (2.641 g/cm$^3$) is close to the value typical for the pure quartz (2.650 g/cm$^3$). The coefficient of variation for this test was very low (CV = 0.14%), which confirms that the lunar regolith simulant is very homogeneous in terms of density (see Table 4).

Among the quartz powders initially tested in terms of particle size distribution, the powder that turned out to be the most similar to the lunar regolith simulant discussed earlier was the one whose PSD plots were also included in Figure 4 above, and the statistical parameters describing the distribution are presented in Table 4. The quartz powder, however, is a bit finer than LRS. It contains particles smaller than 1 μm, the median is 21.5 μm (42% smaller than the median of LRS), average size and mode are about 30 μm,

and maximal size slightly exceeds 150 μm. All these results indicate that the quartz powder has a much larger (1.77× larger) specific surface area than the LRS, which makes LRS a more advantageous mortar micro-filler than the quartz powder described above.

### 4.2. Rheology of Lunar Mortars

As indicated in Table 2, in which the individual phases of the experimental program were presented, the first step in the attempt to develop the composition of lunar mortars was to determine the maximum content of the "micro" fraction in the aggregate.

By using the method of subsequent iterations of increasing substitution of regular aggregate (standard sand) in the standard mortar, it was shown that with an effective superplasticizer (with the water/cement ratio unchanged equal to 0.50), 1/5 (by weight) of the aggregate can be replaced with very fine quartz powder, and a practically identical consistency can be obtained. The flow table test of the standard mortar showed the flow diameter of 206 mm, and for the mortar containing 20% quartz powder (and the superplasticizer in the amount of 1% of the cement mass), 208 mm. In a situation where the test was carried out for a mortar in which 20% of aggregate was replaced with regolith simulant (the same amount of superplasticizer), the flow diameter was even greater—245 mm, which is indicative of the effect of slightly coarser graining and a smaller LRS specific surface area. However, further attempts to increase the amount of micro-filler in the mortar resulted in a significant deterioration of the consistency, and the use of more fluidizing admixture (according to the manufacturer, the maximum recommended dosage in the concrete mix is 1.5%) resulted in the segregation of the mix and visible water separation. An attempt was also made to increase the content of water in the mixture (to maximize the effectiveness of the admixture), which allowed for the introduction of higher amounts of QP/LRS; however, this approach is completely unjustified in the case of lunar composites, where it is advisable to reduce water consumption as much as possible. Therefore, for the mortar with w/c = 0.50 and 1.0% superplasticizer as an initial optimal content of the micro fraction (QP or LRS) in the aggregate was indicated 20% (corresponding with 13.3% of the total mass of the mortar). The selected compositions and test results of mortars carried out in this research stage are presented in Table 5.

**Table 5.** Compositions of mortars with/without micro-filler (QP—quartz powder; LRS—lunar regolith simulant) and flow table test results (D—mean value of the flow diameter measured in two perpendicular directions).

| No | Water [g] | Cement [g] | Aggregate [g] | W/C [g/g] | Admixture [% of c.m.] | QP [% of agg.] | LRS [% of agg.] | D [mm] |
|---|---|---|---|---|---|---|---|---|
| 1 (ref.) | | | | | 0.0 | 0.0 | 0.0 | 206 |
| 2 | 225 | 450 | 1350 | 0.50 | 1.0 | 20.0 | 0.0 | 208 |
| 3 | | | | | 1.0 | 0.0 | 20.0 | 245 |

### 4.3. Rheology of Lunar Micro-Mortars (Series with w/c = 0.265)

When examining the consistency of micro-mortars, which are cement pastes supplemented with an aggregate of the micro fraction of lunar regolith simulant, an attempt was made to reduce water and lower the w/c ratio in relation to that of the abovementioned 0.50 due to a different degree of filling of such mixes. At this stage of the research, the starting point was determining the proportion between water and cement at which the so-called standard consistency (according to EN 196-3 standard) is obtained. For the used cement CEM I 42.5 R the w/c ratio for standard consistency was experimentally determined at the level of 0.265 (Figure 5 shows the results of these tests). This w/c was then used to determine the compositions of the tested lunar micro-mortars with the LRS. Table 6 contains their compositions—LRS was added to the cement paste in increasing amounts from 2% to 6% of the cement mass.

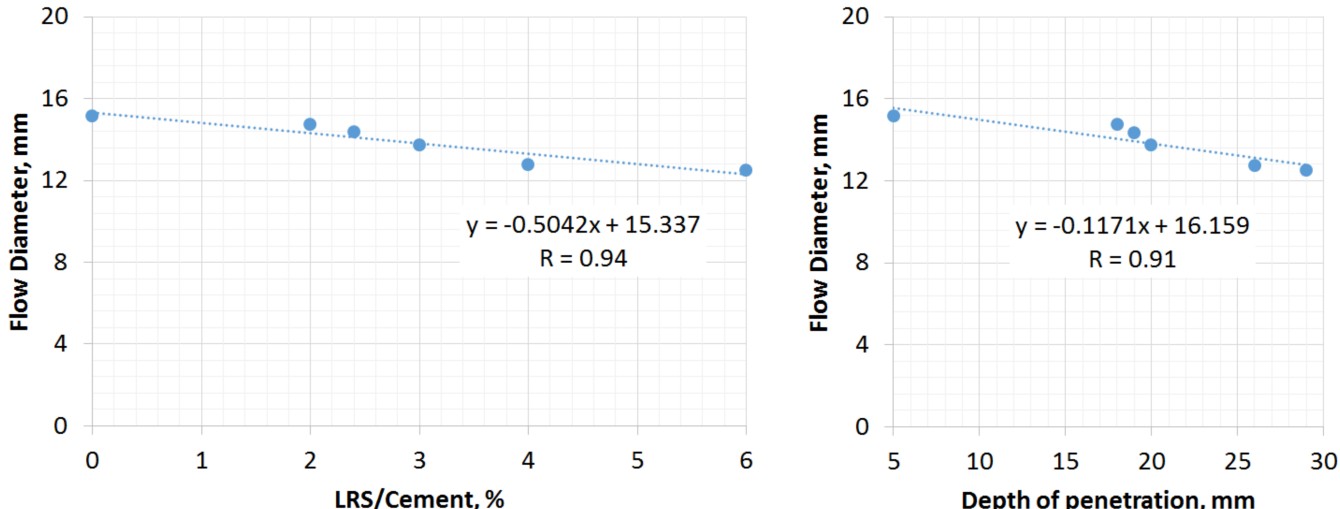

**Figure 5.** Relation between consistency/plasticity of lunar micro-mortars (expressed by flow diameter) and content of lunar regolith simulant (LRS) acc. to the cement mass (**left**) and the relation between flow diameter and depth of penetration in the Vicat apparatus for the same lunar micro-mortars (**right**); all lunar micro-mortars characterized with constant w/c = 0.265.

**Table 6.** Statistical parameters describing the particle size distribution, specific surface area, and density of regolith simulant (LRS) and quartz powder (QP).

| No | Water [g] | Cement [g] | W/C [g/g] | LRS [g] | LRS/Cement [%] | LRS/total mass [%] |
|---|---|---|---|---|---|---|
| 1 | | | | 410 | 82.0 | 35.3 |
| 2 | | | | 500 | 100.0 | 40.0 |
| 3 | 250 | 500 | 0.500 | 520 | 104.0 | 40.9 |
| 4 | | | | 530 | 106.0 | 41.4 |
| 5 | | | | 580 | 116.0 | 43.6 |
| 6 | | | | 600 | 120.0 | 44.4 |

Table 6 contains results of testing the consistency of lunar micro-mortars performed in the Vicat apparatus (measurement of the bolt intender penetration depth in the mix). To assess the plasticity of the micro-mortars, they were also tested on the flow table according to the modified flow table test method, in which a smaller volume of the sample is used (as described in Section 3). The results of the tests are shown in Figure 7.

Based on the analysis of the above data, it is obvious that with increasing LRS content in the micro-mortar mix, the consistency becomes denser. The application of LRS more than 6% of cement mass resulted in the inability to perform the tests—the mixes were too dense to obtain reliable test results. Both tests showed that the relation between the consistency (regardless of whether it was the penetration depth in the Vicat apparatus or the flow diameter) is directly proportional to the content of LRS. Moreover, it indicated that both tests correlated very well. Figure 5 (right) shows the relation between the results obtained with both methods for lunar micro-mortars with identical compositions and the correlation coefficient is high (0.91).

### 4.4. Rheology of Lunar Micro-Mortars (Series with w/c = 0.500)

Significant deterioration of the workability of lunar micro-mortars even at a low LRS content at w/c = 0.265 prompted the authors to study the micro-mortars series with w/c as in the original series, i.e., equal to 0.50. It allowed significant amounts of LRS to be introduced into the micro-mortar mix—from 82% to 120% in relation to the cement mass, which corresponded to 35% to 45% of the total mass of the composites. Further increasing

the LRS content resulted in a situation that the Vicat intender did not penetrate the mix. Since the "penetration depth" according to the EN 196-3 standard is understood as the distance between the intender and the bottom of the Vicat ring, the ring height (40 mm) is the limit value of this "depth." Table 6 contains their compositions. Figure 6 presents the results of the consistency test performed in the Vicat apparatus. This time, a better fit to the empiric data was obtained when the relation between the content of the lunar regolith simulant and the depth of the Vicat intender penetration in the lunar micro-mortar was described using not a linear function but a second-degree polynomial, yielding the correlation coefficient R = 0.96. (For the linear function, the fit was also good but less accurate, and the correlation coefficient was R = 0.94.)

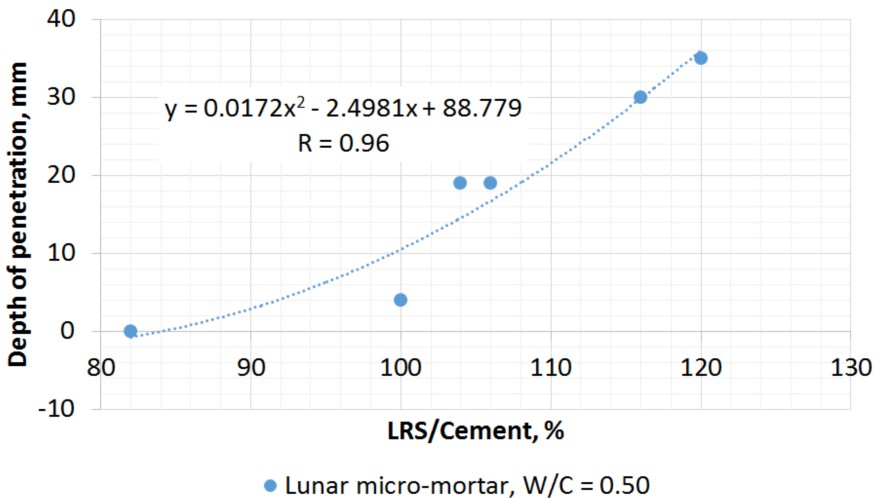

**Figure 6.** The consistency of lunar micro-mortars of w/c = 0.50 expressed by the depth of penetration in the Vicat apparatus (depth is the distance between the intender and the bottom of the Vicat ring; the ring high is 40 mm).

Figure 7 presents the results of the consistency test performed on the flow table—this time, the test was performed in two ways: according to the modified test procedure (smaller volume of specimen, see Section 3) and according to standard test procedure (in accordance with EN 1015-3 standard). For both series of results, regardless of the procedure, an excellent fit to the empirical data was obtained with the use of linear functions (correlation coefficient not lower than R = 0.98). Moreover, the correlation between the results obtained with both methods also proved excellent (R = 0.99). This confirms that the use of a modified flow table test procedure that allows the testing of micro-mortar samples with smaller volumes (equal to the volume of the Vicat ring) provides reliable results and can be used to very precise estimate the result according to the standard flow table test method. For lunar micro-mortars covered by this research, the value obtained by the modified method should be multiplied by a factor of 1.3 to obtain a value corresponding to that of the standard method.

### 4.5. Compressive Strength and Flexural Strength of Lunar Mortar

Mechanical properties of mortars (including lunar mortar) of compositions as in Table 5 are shown in Figure 8. Mortar containing 20% of LRS in aggregate, both in terms of flexural strength and compressive strength, obtained practically identical values as the standard mortar. In the case of analogical mortar containing quartz powder, in both tests, the increase in strength was noted ($\Delta f_b$ = 2.6 MPa, which corresponds to a 38% increase; $\Delta f_c$ = 9.8 MPa, which corresponds to a 20% increase). This can be explained by the finer grading of this micro-filler and potential better packing of its particles between the grains of sand and better sealing the microstructure of the mortar.

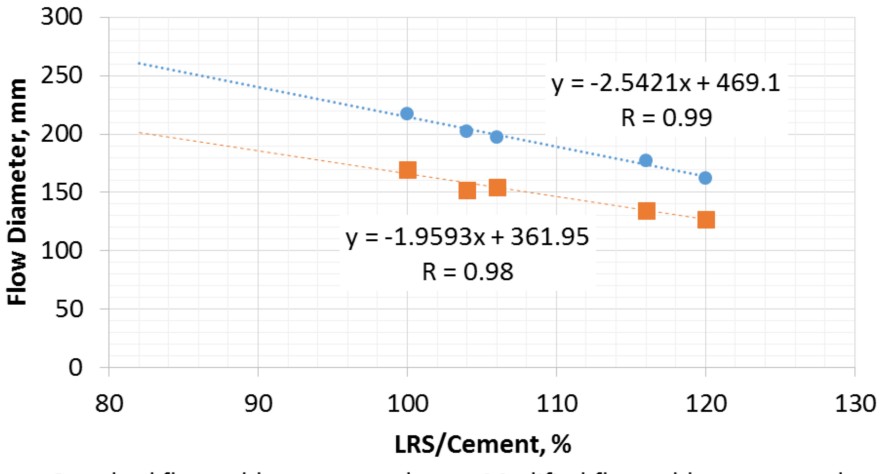

**Figure 7.** The consistency of lunar micro-mortars of w/c = 0.50 expressed by flow diameter (test performed acc. to the modified test procedure and standard test procedure in conformity with EN 1015-3).

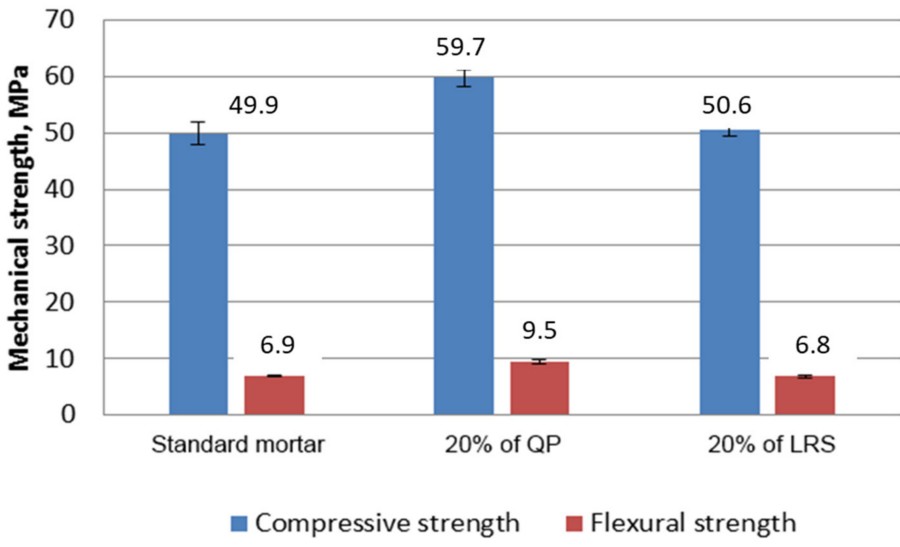

**Figure 8.** Compressive strength and flexural strength of tested mortars, including standard mortar with CEMI 42.5R and mortars containing 20% of quartz powder (QP) or lunar regolith simulant (LRS) in the total mass of aggregate.

## 5. Additional Considerations in the Context of Cement-Based Lunar Materials

Lunar outpost construction would require innovative solutions on almost all levels. With extreme external conditions on the Moon, namely, lack of atmosphere and high cyclic temperature changes, the continuous supply of thermal energy and power is a crucial issue as they are necessary to create and maintain conditions in which any building object could be constructed. For most locations on the lunar surface, darkness lasts for periods of about 350 h; thus, it is a great challenge for the solar photovoltaic cells and radioisotope thermoelectric generators to provide enough energy [33]. The lack of atmosphere on the Moon has an additional crucial impact on the design of any lunar structure—ionizing radiation—which impacts both future residents of the constructed buildings and the dura-bility of used construction materials. Due to the missing magnetosphere and atmosphere, particles of galactic and solar origin reach the surface of the Moon unattenuated. Any construction on the Moon, including the ones built from regolith, is required to reduce the exposure to safe levels for prolonged human presence on the Moon [34] as well as protect

it against meteoroids impacts with velocities that vary from 2.4 km/s to 72 km/s and weighing from less than 1 kg to over 5 tons [35]. When designing the lunar construction, one needs to take into consideration all the abovementioned factors.

The lunar regolith morphology resulted from the continuous impact of meteoroids and the bombardment of the lunar surface by charged particles from the Sun [8]. It has low thermal conductivity, and therefore, the thicker its layer is, the larger the reduction in the temperature variations is [35]. Additionally, from the perspective of radiation health, a regolith layer of 2–3 m thickness provides radiation shielding of about 40 g/cm$^2$, whereas Earth's atmosphere provides a shielding equivalent of 100 g/cm$^2$. Rare, large solar flares require shelters with shielding of at least 700 g/cm$^2$ [35]. By using lunar regolith as an aggregate for mortars or concretes, it is possible to take advantage of those properties and increase the shielding properties of the new construction composites.

While the impact of the lunar environment on the solidification of cement and its properties is nascent, a great deal can be learned from studies conducted on Earth utilizing JSC-1A regolith simulant. JSC-1A can be a feasible precursor for alkali activation, the resulting product of which has been termed Lunamer [36]. It achieved 16 MPa in compressive strength when cast with conventional methods and provides good radiation shielding and thermal insulation [37]. Alkali-activated materials present a relatively new group of construction materials and therefore require an additional significant amount of research in both terrestrial and extraterrestrial settings [38].

The microgravity conditions present on the Moon result in an increase in porosity and pore sizes in hydrated cement pastes [39]. Admittedly, using OPC (and certainly manufacturing it) on the lunar surface is nontrivial; the hurdle is the environment. Despite the high cost associated with transporting cargo to space, shipping small amounts of OPC to the Moon could be an alternative to initiate the construction of a lunar outpost until the crew is ready to process the in situ resources to manufacture materials.

In order to increase the overall properties of extraterrestrial cement composites, additive manufacturing using in situ resources (ISRU) is being developed to facilitate future off-planet habitation by enabling the development of infrastructure and other objects without the need for dedicated manufacturing facilities or distant transport of materials [40]. Laser sintering, direct printing, and powder binder jet printing have already been proposed as potential solutions in developing a lunar habitat [6]. Each of these methodologies has been tested utilizing various regolith simulants. With the use of laser sintering bricks, gears, nuts, and other components have been produced [41], exhibiting a wide range of compressive strengths (5–420 MPa) [42]. Powder binder jet printing was developed to manufacture structural elements of habitat and the feasibility of printing and curing the cementitious binder in vacuum conditions has already been validated [43].

All those methods require the consistency of manufactured cement mortar or concrete mix to be at a satisfactory level allowing for the use of any given technology. Water is a scarce lunar resource and the water demand for construction materials production competes with the demand for other crucial water applications. Therefore, there is a benefit in developing methods to recover water from these cementitious binders while producing a material with specific mechanical parameters and sufficient durability. Lunar regolith-based magnesium oxychloride (MOC) cement composites are promising candidates for water recovery [40]. These types of cement are based on the reaction between MgO and an MgCl$_2$ solution, enabling gel phases to be produced that lead to a high-compressive-strength binder that can incorporate a large amount of filler material, such as lunar regolith [44].

## 6. Conclusions

In the conducted research, the influence of powder simulating the granulometry and morphology of lunar regolith on the rheology of lunar micro-mortars and mortars was tested. The microscopic observations confirmed the sharp-edged morphology of LRS particles, characteristic of the lunar soil, which unfortunately affects negatively the

consistency of mixes of cementitious composites with these fillers. The determining particle size distribution of LRS showed strong similarities with the real lunar regolith data. The used LRS characterized by a density of 2.641 g/cm$^3$, i.e., a very similar to that of quartz (2.650 g/cm$^3$), which allowed for the mass substitution of quartz aggregate in mortars; however, due to the much finer granulation in comparison to standard sand, it was decided that the substitution would be considered for a composite containing both quartz sand and quartz micro-filler similar to the LRS in terms of PSD.

As presented in the Results Section, very fine LRS of much more developed specific surface area (thus also much higher water demand ratio) than standard sand had a negative impact on the rheological properties of cementitious composites; therefore, it required the use of a very effective superplasticizer. Ultimately, it was possible to obtain a mortar mix in which 20% (by weight) of the aggregate was replaced with LRS, which is characterized by very similar (or even slightly better) plasticity in comparison to the standard mortar. Moreover, the tests of the mechanical properties carried out after 28 days showed that the mortar modified in this way was characterized by identical values of flexural strength and compressive strength as the standard mortar (respectively 7 MPa and 50 MPa). Taking into consideration the lunar environment, characterized by much lower gravitational pull than Earth's, those results confirm that mechanical properties of regolith-based cement composites are sufficient for lunar construction purposes [37].

In order to better investigate the relation between the cement paste phase and LRS, and considering the need to create cement composites using only LRS as fillers, based on the results of two series of lunar micro-mortars with a w/c ratio that differs almost twice (0.265 and 0.500), the dependencies were developed in the form of very well fitted to empiric data functions describing the LRS/Cement–mix consistency relation.

Taking into account that nonstandard concreting of the mortar mixture by 3D printing is a potential solution in the Moon conditions, it was shown that it was possible to obtain a micro-mortar consistency suitable for this technology. Considering that the flow values obtained by the standard method on the flow table (see Figure 8) were within the range recommended by scientists working on 3D printing technology (compared to the recommended ranges described in the last part of Section 3, i.e., from 180 to 240 mm), it can be concluded that the developed micro-mortars meet those requirements. However, the next stage of optimization research should include buildability time.

The appropriate consistency of cementitious composites enables the easier production of cement-based products. Performed research showed that in order to consider the use of cement composites on the Moon, the issue of regolith's impact on its consistency needs to be addressed. The obtained results allowed to formulate a conclusion that the use of the lunar regolith in mortars and micro-mortars has potential, but their qualitative and quantitative compositions should be very precisely selected, especially when trying to develop compositions of lunar mortars with much lower water contents. The new generation superplasticizers (e.g., with hyperbranched polymers [45,46] or polyphosphate comb polymers [47]) of highly improved efficiency in fluidizing could prove useful in increasing lunar regolith content in cementitious composites while maintaining consistency and w/c at reasonable levels.

**Author Contributions:** Conceptualization, J.J.S. and P.W.; resources, J.J.S.; selected data was obtained during the completion of the diploma thesis by P. Kisielewski under the supervision of J.J.S.; writing—original draft preparation, J.J.S. and P.W. and M.K.; writing—review and editing, J.J.S., P.W. and M.K. All authors have read and agreed to the published version of the manuscript.

**Funding:** This research received no external funding.

**Institutional Review Board Statement:** Not applicable.

**Informed Consent Statement:** Not applicable.

**Acknowledgments:** The authors would like to express their gratitude to Paweł Woliński, and Karol Seweryn, from Space Research Center of the Polish Academy of Sciences for their support in obtaining LRS.

**Conflicts of Interest:** The authors declare no conflict of interest.

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
