# Peer review of "Rheological Properties of Lunar Mortars"

_applsci, doi:10.3390/app11156961_

Round 1

Reviewer 1 Report

Article: applsci-1251601

Article entitled “Rheological properties of lunar mortars” is discussing the mortar made with lunar aggregate. Most articles discuss the basic experimental tests and its result discussion using the lunar regolith simulant. Authors discuss mostly the experimental results and they didn’t discuss about the scientific findings and its relative discussion. Authors are requested to explain the methodology of work through a figure that is more efficient. Some sentences are too lengthy and some are grammatical mistakes. Kindly check it. Authors are requested to add a few more literature about the already available literature on lunar concrete/mortar with its properties and try to explain, what is new in this study? LRS being fine material, it is more susceptible to shrinkage and creep and the author wants to address this. Authors are requested to compare the chemical properties of lunar regolith, LRS and quartz powder. Authors are requested to rewrite the abstract, to make sure that the whole content of the article should replicate in the abstract. Authors are requested to check the nomenclature of the article throughout. Articles lags in references to literature throughout especially in the result and discussion part. Conclusion is too lengthy and it should be shortened, preferably, bulletin.

                Due to lack of discussion about scientific reasons behind experimental work. I suggest a major revision in the article. In order to improve the quality of article, I requested to address the few question below apart from those I added in pdf document

  1. Include the experiments and overall output
  2. Keywords should replicate the content in the abstract
  3. As the author describes in the introduction part, when concrete on the moon is exposed to solar radiation, there will be a drastic effect on the mineral composition on it. How to rectify it?
  4. One of the factors that governs W/C ratio of mortar mix is fineness modulus of aggregate. Authors are requested to report the fineness modulus value of LRS. Authors are requested to report several other parameters like specific gravity, water absorption, etc of LRS.
  5. Line 105-107: It is stated that “lunar regolith includes particles smaller than 10 μm, of which about 90% (by weight) are particles smaller than 1 μm [19]; although some particles can 107 be even as small as 10 nm” But in the line 346-347, it is stated that “The real lunar regolith consists in about 90% of particles larger than 10 μm”. Both lines are controversial. Kindly explain.
  6. It is well known that the determination of consistency of standard cement as from Figure 5. What is new in this?

Author Response

Dear Reviewer,
On behalf of myself and the co-authors, I would like to thank you for the valuable remarks and indicating the necessary corrections. We have included all the indicated editorial corrections to the manuscript. Moreover, below are additional explanations to your remarks and doubts given in the first, general part of the review, as well as remarks given in additional PDF file.  And in the following, we answer the numbered questions/remarks.

We hope that everything has been clarified and explained and our paper quality is now good enough for publication.

Yours Sincerely,
Joanna J. Sokołowska, Ph.D.

Answers to remarks in general part of the review and additional PDF

Remark: Authors are requested to explain the methodology of work through a figure that is more efficient.

The methods used to obtained results are commonly known and used in the whole Europe, and the authors avoided a more detailed explanation of the research procedures so that they would not be accused of describing obvious matters. The methodology was explained in the manner that is generally accepted:

  • If there was used a particular method described in the European standard, the number of the standard was given (however always some details had been given like the used equipment or machines). This is exactly what was done in case of density test in Le Chatelier volumenometer, consistence test using Vicat apparatus and flow table and mechanical properties.
  • If the standard method was changed during the test, then the authors described the details of the changes in the procedure. This happened when the sample volume was changed during the test on the flow table.
  • If the method was not described in the standard, then the parameters needed for the test and the method principle were given. This was the case with the examination of granulometry in a laser granulometer and taking pictures of the samples.
    The laser granumometer works on the basis of the Mie solution, but a detailed description of exactly which way (through which lenses, etc.) the laser beam passes and the parameters of this beam are technical information that is too detailed and would not contribute to the analysis of the test results. Also with regard to the photographs taken, it was found that there is no need to describe any more details as the description of taking process of taking a picture is also of no value for image analysis.

It is not very clear, therefore, what better to describe the methodology so as not to go into unnecessary details of the construction of devices or to include descriptions of very well-known studies that – as mentioned – might seem too obvious for a scientific article. If, however, the reviewer requires specific details, we ask you to indicate which method is too little known and should be clarified.

Remark: Authors are requested to add a few more literature about the already available literature on lunar concrete/mortar with its properties and try to explain, what is new in this study?

As mentioned in the paper, so far there has been no practical sense in doing lunar mortar tests with cement and water, because bringing water to the Moon was not taken into account, and water was not expected to be obtained on the Moon. So far, the focus has been on concretes with polymer or sulfur binders. Such composites were mentioned in the paper, but not much attention was paid to them, because they have practically no relation to the composite studied by the authors (completely different material composition, setting and curing, and properties). The approach of mixing lunar regolith with cement and water is virtually a new approach, which was inspired by recent NASA reports of finding frozen water (so called “water ice”) on the Moon. Therefore, the authors did not have a chance to cite many scientific papers on this topic or compare the results.

Remark: LRS being fine material, it is more susceptible to shrinkage and creep and the author wants to address this

LRS seems similar to quartz powder in terms of granulation. It is even less fine than quartz powder, that was used in some concrete-like materials (eg. aerated concretes) for years, so one can accept similar effect when it comes shrinkage and creep. Moreover, these properties were not planned as the subject of the paper. The paper was focused on developing suitably liquid mixes. The shrinkage and creep long-term studies could be considered the next stage of research and the topic of a separate article. Thank you for this valuable remark and idea.

Remark: Authors are requested to compare the chemical properties of lunar regolith, LRS and quartz powder.

This absolutely is unnecessary for the study because, as clearly stated, the LRS was intended to simulate the granulation of real regolith, not the chemical composition. And it was the aspect of the influence of particle size on consistence that was taken into account. No chemical reactions, no hydration nor hydrolysis was analyzed.

Besides - as explained in section 1.3 .: "Polish lunar regolith simulant is a mix of components appearing in Polish geological conditions  and/or commercial components produced by local raw mineral producers, including mechanically crushed quartz, granite sand and grit." So the chemical composition of LRS was not much different than regular concrete or mortar aggregate and included in general pure silica or compounds of quartz present in Polish granite – which also was explained in section 4.1: “tested material was a simulant obtained by grinding rocks rich in quartz (silica, granites from Polish deposits, which are mainly made of plagioclase, quartz, biotite and hornblende”.

But once again it should be emphasized that the premise was to simulate the effect of granulation and LRS grain shape on the mortars consistence.

Remark from the PDF: How author had done this SEM micrograph? (line 115)

The mentioned SEM micrograph showed on the Fig. 2 is cited from the paper listed in the references on the [20] position, which was clearly said in the Fig. 2 caption. But the citation has also been added in the indicated place (line 115).

Remark from the PDF: How superplasticizer dosage is fixed as one percent of mass of cement?

The dosage of the plasticizer and other concrete admixtures is always in the amount up to 5% in relation to the cement mass. This is a standard approach to concrete modification and is described in many standards (including EN 206). And it is always described as such. For example, if the mix uses 300 kg of cement per 1 m3 of mix, then the admixture is added in the amount of for example 1% of the cement mass, ie 0.01 * 300 kg = 3 kg of admixture per 1 m3 of mix.

Remarks from the PDF: Remove the sentence (lines 992-393)

The marked part of sentence saying “according to the manufacturer, the maximum recommended dosage in concrete mix is 1.5%”  is very necessary from the point of view of concrete technology. According to the standards for cement modification, dosing of the plasticizing admixture can be up to max. 5% of cement mass. The information that the maximum recommended dosage is 1.5% indicates the effectiveness of this modifier. Such information is practically the only way to assess the effectiveness of the admixture, because manufacturers do not provide the admixture chemical compositions, which information is a company secret. This is why authors cannot remove this sentence.

Remarks from the PDF: The reviewer asked about meaning of the abbreviations used in the paper:

The authors explain, although these are not authors own abbreviations, but abbreviations (partly originating from the Latin) that are officially accepted to use in English:

  1. circa about (line 47)
    i.a. –
    inter alia, meaning 'among others' (line 217)
    acc. – according to (line 410)

Answers to numbered remarks of the Reviewer.

  1. Include the experiments and overall output

As for the scope of the experiment and the sequence of testing, it has been thoroughly discussed in the section 2- in the text and additionally in Table 2.
As for experiments description, it was already explained, that the description was made following the convention typical for the research carried out on standard procedures. Moreover, authors did not elaborate on the descriptions to avoid writing things that are obvious and very basic.

  1. Keywords should replicate the content in the abstract

Keywords were revised and changed as recommended.

  1. As the author describes in the introduction part, when concrete on the moon is exposed to solar radiation, there will be a drastic effect on the mineral composition on it. How to rectify it?

This hazard does not apply to cement concrete or mortars. Radiation is a destructive factor for polymer concretes and, of course, a threat to human health. Cement concrete and mortars, on the other hand, is an effective barrier against radiation.

  1. One of the factors that governs W/C ratio of mortar mix is fineness modulus of aggregate. Authors are requested to report the fineness modulus value of LRS. Authors are requested to report several other parameters like specific gravity, water absorption, etc of LRS.

Testing in a laser granulometer gives results in the range 0.001 µm to 600 µm.The curve is built from many dozens of measurements, not as in the case of a sieve test of coarser aggregates – several contents.For the population of results (within one, but very complex test), numerous distribution statistics are determined (which were not presented in the paper, so as not to obstruct the analysis), however, fineness modulus is not computed.Moreover, it should be remembered that the same value of fineness modulus may therefore be obtained from several different particle size distributions. Also the fineness modulus definition is different depending on national guidelines and recommendations and is not used as common criterion for very fine aggregates characterization.

Therefore, during the PSD analysis, the focus was on specific parameters of individual distributions, such as mode or median, because they were considered more informative evaluation criteria in the analyzed case of micro-fraction.

  1. Line 105-107: It is stated that “lunar regolith includes particles smaller than 10 μm, of which about 90% (by weight) are particles smaller than 1 μm [19]; although some particles can 107 be even as small as 10 nm” But in the line 346-347, it is stated that “The real lunar regolith consists in about 90% of particles larger than 10 μm”. Both lines are controversial. Kindly explain.

There is no contradiction here.
“The real lunar regolith consists in about 90% of particles larger than 10 μm.”
That means it consists in 90% of particles larger than 10 μm and 10% of particles smaller than 10 μm.

“lunar regolith includes particles smaller than 10 μm, of which about 90% (by weight) are particles smaller than 1 μm
When we focus on those smaller ones (because they are more important when it comes to the water demand ratio), we discover that 90% of those smaller than 10 microns are the particles smaller than 1 μm.

  1. It is well known that the determination of consistency of standard cement as from Figure 5. What is new in this?

Standard consistence is a characteristic of a particular cement. It is not declared or required in the standard. It is different not only for different class (32.5, 42.5 and 52.5) and types of cements (CEM I, CEM II, etc.), but also different for cements of the same class/type produced by different manufacturers and even for different batches of cements produced by the same cement plant.

From the point of view of concrete technologists it is an absolutely necessary parameter and practically the only parameter that says anything about the potential rheology of mixes with a given cement. The knowledge of the real w/c corresponding to standard consistence was the starting point for further tests done in the research taking into account the consistence.

Reviewer 2 Report

The manuscript should be written more clear and in detail, following special justified objectives. As an example how the author achieved the below result:

"The obtained results made it possible to develop preliminary compositions for “lunar mortars” (possible to apply in e.g. 3D concrete printing) and to prepare, test and evaluate mortars properties in comparison to traditional quartz mortars (under the conditions of the Earth laboratory)."

What are the criteria for possibility to apply the research outcomes to 3D concrete.

Author Response

Dear Reviewer,
On behalf of myself and the co-authors, I would like to thank you for the valuable remarks and indicating the lacking issues.

In response to the issue raised in the review regarding 3D printing, the paper was supplemented with recommendations on the workability of cementitious mixes (end of section 3). The relevant research papers have been cited and included in references. Additionally, in the conclusions section the obtained results have compared with these recommendations.

Additionally, we have introduced language and editing corrections.

We hope that everything has been clarified and explained and our paper quality is now good enough for publication.

Yours Sincerely,
Joanna J. Sokołowska, Ph.D.

Reviewer 3 Report

The research presented in the article entitled “Rheological properties of lunar mortars” addresses to the use of a lunar regolith simulant as aggregate in lunar concrete. The study analyzes the erosive effect of dusty regolith fractions on the moving parts of lunar landers and other mechanical equipment. The article ensures very complex and useful data about an innovative material for using in an very different environment than usual. It is a very interesting study, with a high focus on the future possibilities.

The Abstract presents a clear and comprehensive statement of the study described in the paper.

The Introduction and the entire paper provide enough literature references about the article’s subject and for the analysis performing.

The materials and methods are in detail described. The results are in detail presented, clearly highlighted and discussed, and the conclusions summarize them properly.

Round 2

Reviewer 1 Report

The article has been considerably improved, so I recommend to be accepted in the present form.
I only want to make two comments about it:
1. For determining the dosage of superplasticier.. we normally do a marshall cone experiment. But other of this article says that they had used directly 1.5% as per recommendation of the manufacturer.
2. Chemical composition of the new ingredients is required for every inspection. But the author said that there is no.